# Tics and Emotions

**DOI:** 10.3390/brainsci12020242

**Published:** 2022-02-10

**Authors:** Gerry Leisman, Dana Sheldon

**Affiliations:** 1Movement and Cognition Laboratory, Department of Physical Therapy, University of Haifa, Haifa 3498838, Israel; 2Department of Clinical Neurophysiology, Institute for Neurology and Neurosurgery, Universidad de la Ciencias Médicas, Havana 10400, Cuba; 3Department of Cognitive Neuroscience, George Washington University, Washington, DC 20052, USA; danalsheldon@gmail.com

**Keywords:** tics, emotions, basal ganglia, Tourette syndrome, dopamine, HPA axis, premonitory sensory phenomena

## Abstract

Tics can be associated with neurological disorders and are thought to be the result of dysfunctional basal ganglia pathways. In Tourette Syndrome (TS), excess dopamine in the striatum is thought to excite the thalamo-cortical circuits, producing tics. When external stressors activate the hypothalamic-pituitary-adrenal (HPA) axis, more dopamine is produced, furthering the excitation of tic-producing pathways. Emotional processing structures in the limbic are also activated during tics, providing further evidence of a possible emotional component in motor ticking behaviors. The purpose of this review is to better understand the relationship between emotional states and ticking behavior. We found support for the notion that premonitory sensory phenomena (PSP), sensory stimulation, and other environmental stressors that impact the HPA axis can influence tics through dopaminergic neurotransmission. Dopamine plays a vital role in cognition and motor control and is an important neurotransmitter in the pathophysiology of other disorders such as obsessive–compulsive disorder (OCD) and attention deficit hyperactivity disorder (ADHD), which tend to be comorbid with ticking disorders and are thought to use similar pathways. It is concluded that there is an emotional component to ticking behaviors. Emotions primarily involving anxiety, tension, stress, and frustration have been associated with exacerbated tics, with PSP contributing to these feelings.

## 1. Introduction

Tics are phenomena or symptoms associated with dysfunction in the pathways of the basal ganglia, which are largely of childhood onset. While many symptoms such as tremors and spasms are all concerned with basal ganglia dysfunction and can look similar to ticking, it is important to provide a specific definition. Tics can be characterized by premonitory urges, or premonitory sensory phenomena (PSP) that encompass a variety of sensations, including focal tension, burning, and itching [1]. Many individuals have reported that these urges are worse than the tics themselves and have stated that their tics are either a partial or fully voluntary response to PSP [2]. Additionally, tics are not task-specific, while other conditions such as stuttering and dystonia include basal ganglia dysfunction; it is only when specific tasks are performed (e.g., talking and walking) that the behavior becomes relevant [3]. In other words, tics are associated with PSPs, and occur independently of any task. Additionally, stereotypies and tics can be mistaken for one another; there are characteristics that can clarify the differences between the two. Stereotypies have an average onset age of 3 years, while tics will typically emerge between 5–7 years old [4]. Additionally, stereotypies include constant and fixed movement patterns, rhythmic and prolonged or continuous movements, are not associated with premonitory urges, and can be suppressed with distractions [5]. Conversely, tics are associated with quick and sudden movements and the urge to tic is usually preceded by a premonitory urge(s) [4]. Tics are also most common in the eyes, head, and shoulders, while stereotypies usually occur in the mouth, hands, and arms, sometimes involving the whole body [4]. With the features of tics established, we will examine the role of the basal ganglia and its relationship to ticking behaviors, with an emphasis on the relationship between emotions and ticking behaviors.

The basal ganglia consists of a small yet complex group of subcortical nuclei located deep within the cerebral hemispheres at the base of the forebrain. The primary nuclei of the basal ganglia (the striatum, globus pallidus, subthalamic nucleus, and substantia nigra) are significant with regard to motor control, emotion, and cognition. It is imperative to examine the interconnections between these functions to better understand the nature of motor and verbal tics. The basal ganglia-cerebellar-thalamo-cortical system provides a mechanism for tic production and can also aid in comprehending the connection between emotions and tics.

The basal ganglia, cerebellum, and thalamus are a densely interconnected network that is thought to influence many of the same areas of the cerebral cortex. Two routes of information processing in this network have been demonstrated: the direct and the indirect pathways. In the direct pathway, medium spiny neurons (MSNs) in the striatum project directly to the output nuclei, which include the internal globus pallidus (GPi) and the substantia nigra pars reticulata (SNr). The indirect pathway takes a longer approach, where MSNs project onto the external globus pallidus (GPe), which then projects onto the GPi and SNr [6]. The GPe has also been observed to project to both the motor and non-motor regions of the dentate nucleus when rabies virus was injected into different spatial regions of the GPe of macaque monkeys, which resulted in transneuronal transport to neurons in the cerebellum [6,7]. Therefore, it can be concluded that cerebellar output targets medium spiny neurons in the indirect pathway of the basal ganglia-cerebellar-thalamo-cortical circuit [6]. This is an important finding, as abnormalities at one end of the network could spread throughout the entire system, affecting activity in other places along the pathway. Because the amygdala, hypothalamic-pituitary-adrenal (HPA) axis, and other aspects associated with emotions play into this network, it is thought that emotions can influence ticking behaviors.

It has been known for some time that emotional variables influence tic behavior. Bornstein and colleagues [8] reported that the majority of their participants (98.2 percent) related that anxiety or perceived stress exacerbated their tic behaviors. Their results were confirmed by Eapen and associates [9] and by Silva and colleagues [10]. Findley and associates [11] provided additional support for the relation between stressful life events and tics. They noted that, in 32 children with TS and OCD, significantly greater stressful life events were reported compared with matched controls. Hoekstra and associates [12], in a longitudinal study, found significant correlations between negative life events and tic severity. O’Connor et al. [13,14] found that frustration was an emotional state that was mostly related with high-risk and ticking behaviors. The literature generally supports the notion that anxiety, stressors, and frustration are significantly related to increases in the frequency and intensity of tic behavior, with a review of significant findings reported by Conelea and Woods [15].

Tourette syndrome (TS) is the most commonly diagnosed tic disorder, but not every tic disorder is indicative of TS. Many tic disorders use similar pathways in the basal ganglia circuitry to produce ticking behaviors, but the presence of tics should not result in an immediate assumption of TS. In TS, tics are associated with abnormalities in the basal ganglia, particularly with the striatal GABAergic networks, leading to an excess of striatal dopamine [16,17]. 

Changes in striatal dopamine release can induce tic production due to “a focal excitatory abnormality in the striatum that causes undesired disinhibition of thalamo-cortical circuits, whose effect is the production of tics” [17]. When there is a further influx of dopamine to an already overabundant supply in the striatum, tic production can be exacerbated. For example, the HPA axis responds to acute stress by producing the corticotropin-releasing hormone (CRTH), which in turn promotes the release of the adrenocorticotropic hormone (ACTH) from the pituitary gland and cortisol from the adrenal cortex [18]. While these actions are essential for coping with stress, it is thought that they are mediated by dopamine release [18]. This paper will further delve into the specifics of how stress and other emotions affect tic production, as well as how dysfunction in the basal ganglia circuitry can lead to other neurological and behavioral deficits.

The emotional components of tics may also have an effect on immunity and health. HPA axis activation due to psychological stress can be evoked by infections and injuries, and it is suggested that the influence of stress on ticking behaviors is related to the influence of the immune system on tic expression [19]. Because dopamine is a major neurotransmitter that is vital to HPA axis activation, any changes in dopaminergic neurotransmission can have an effect on stress, tic expression, and immunity [19]. Martino et al. [19,20] developed a model that described how an “enhanced autoimmune response might contribute to the onset or worsening of tics via dopaminergic neurotransmission” [19,20]. This model explains that psychosocial stress can indirectly contribute to increased dopamine release from the sympathetic nervous system, which can lead to an enhanced autoimmune response [20]. Additionally, other studies observed that plasma dopamine levels increased with reports of stress: this is the primary source of dopamine for T-cells, which help regulate immune response [21].

It is also possible that the increased HPA axis activation observed in TS individuals could be a consequence of elevated inflammatory responses to other infections [19].

Effector molecules such as cytokines modify the activity of cells involved in immunological reactions. Leckman and colleagues measured plasma levels of multiple cytokines in TS individuals and healthy controls, “reporting increased baseline levels of tumor necrosis factor-α (TNF-α) and interleukin-12 (IL-12)” [22]. Generally, the authors found that there was an increase in these two cytokines, in addition to many others, with increased ticking symptoms. Because TNF-α and IL-12 are involved in inflammatory responses, it is thought that the immune response may be overactive in people with TS [20].

In addition to an increase in cytokines being associated with an overactive immune response, it is also thought that streptococcal infections and autoantibodies can play a role in the immune system of individuals with tic disorders. A study by Leslie and colleagues found that individuals with newly diagnosed OCD, TS, or tic disorders were more likely to have been diagnosed with a streptococcal infection the year before tic onset, providing evidence that streptococcal infections may later induce ticking behaviors [23]. Additionally, Group A streptococcal bacteria are thought to include T and B lymphocytes, whose antibodies or cytokines cross the blood–brain barrier to alter neurotransmission and produce chorea [20,24]. This is still being tested in TS and other ticking disorders, but the role of antibodies is still controversial, and antibody detection methods are still being perfected. However, it has been observed that individuals with SC have auto-antibodies that bind to the cytoplasmic proteins of neurons, particularly in the basal ganglia [25,26], which are only available after damage or disruption [20]. Figure 1 illustrates the hypothesized pathways potentially involved in the relationship between autoimmune compromise and ticking behavior.

Excess dopamine in the basal ganglia circuitry is thought to produce the symptoms of tic disorders, overexciting the circuits and producing excessive motor output. Dopamine itself is an essential neurotransmitter in the central nervous system, and plays a role in motor control and cognitive function through its interaction with dopamine receptors (DRs) D1-D5 [27]. Not only are DRs prevalent in the basal ganglia-thalamo-cortical-circuit, but they are also expressed on peripheral blood lymphocytes (PBL) in humans, whose function is thought to be impaired with disorders involving the CNS dopaminergic circuits [27,28,29,30,31]. While many studies indicate that this is true for diseases such as schizophrenia, Alzheimer’s, Parkinson’s, and major depression, Ferrari and colleagues sought to understand if this was also true for TS individuals. They found an upregulation in DRD5 mRNA expression in TS individuals compared to normal controls, which is significant because DRD5 is involved in antioxidant and anti-hypertensive responses [27,32]. Additionally, Ferrari and colleagues found that the upregulation of DRD5 by excess dopamine is correlated with an extreme reduction in the function of the CD4+ and CD25+ regulatory T-cells. The decreased frequency in these regulatory T-cells is a common finding in autoimmune disease [27,33].

## 2. Anatomical Circuitry and Tic Behavior

### 2.1. Motor Aspects of Tics

The nuclei that make up the basal ganglia, as shown in Figure 2, include the striatum (which is further divided into the putamen and caudate nuclei), subthalamic nucleus (STN), the internal and external globus pallidus (GPi and GPe, respectively), the substantia nigra pars compacta (SNc) and pars reticulata (SNr). Projections from the cortex and the thalamus then project striatum and STN (through the direct and indirect pathways described above) to the basal ganglia output structures, the GPi and SNr. Basal ganglia activity is modulated primarily through the neurotransmitter, GABA (since the projection neurons in the striatum, GPe, GPi, and SNr are all GABAergic) but can also be influenced through dopaminergic projections from the SNc [34].

A direct link has been observed between basal ganglia dysfunction and motor tics. It has been found that tics are primarily caused by abnormalities in the GABAergic striatal networks, leading to an excess of striatal dopamine. When the GABAergic antagonist bicuculline was injected into the putamen (sensorimotor part of the striatum) of monkeys, striatal inhibition was increased, which was associated with abnormalities in dopamine release, causing motor tics [16,35]. During ticking behavior, neuronal activity was recorded in the basal ganglia, cerebellum, and primary motor cortex (M1) in order to further investigate the relationship between the structures. While the results confirmed basal ganglia dysfunction to be the ultimate cause related to motor tics. The work of [35] also found that motor tics were associated with the enhanced activity of the M1 and cerebellum, implying that these structures may act together to produce motor tics [35]. Due to the fact that the activity of the M1 and cerebellum overlapped and followed the basal ganglia activity, it is thought that the basal ganglia could influence the excitation of these structures, possibly influencing tic behavior [16].

The basal ganglia system can be subdivided into three distinct areas consisting of associative, sensorimotor, and limbic regions [36]. As is clear from the information above, motor tics originate in the sensorimotor area, which consists of the putamen in the striatum. However, vocal tics are thought to emerge from the limbic region in the basal ganglia, which includes the ventral striatum projecting to the anterior cingulate cortex (ACC) and medial orbitofrontal cortex [37]. This limbic territory is thought to be involved in emotional and motivational processing. It was observed that the ACC and M1 display readiness potentials associated with motor planning precede voluntary utterances, emphasizing the importance of the ACC and M1 in vocal control [38]. McCairn and colleagues [37] injected bicululline into the nucleus accumbens (NAc, a central component of the limbic striatum), which successfully produced vocal tics that were described as ‘grunts’. Following the disinhibition of the NAc, it was found that cerebral blood flow bilaterally increased in the ACC, amygdala, and hippocampus, indicating that these structures are involved in vocal tics [37]. PET imaging revealed activation in the ACC, amygdala, and hippocampus during vocal tics, confirming the involvement of the limbic region [37]. Because these same structures are involved in emotional processing, we propose that tics have an emotional component that can be explained through interconnections between the basal ganglia-cerebellar-thalamo-cortical system, the HPA axis, hippocampus, and amygdala.

The amygdala has direct connections to the striatum of the basal ganglia, in addition to areas of the brain involved in fear and anxiety. The striatum funnels information from the limbic system to the motor system, demonstrated by the convergence of amygdala and hippocampal inputs in this area [39]. Interactions between the amygdala, striatum, and prefrontal cortex coordinate emotional, cognitive, and motor behavior: the amygdala, which processes emotions, projects to the striatum, which allows these signals to affect motor behavior [39,40]. The prefrontal cortex then “decides” on whether or not to act on this activity [39].

In line with this, Butler et al. found that activity in the amygdala was only present in the initial period of a threat [41]. This same study also found increased activity bilaterally in the putamen, thalamus, and caudate nucleus in response to a fearful situation. Additionally, Jackson and Moghaddam [42] found that amygdala output can directly increase dopamine signaling in the striatum. It is thought that the abnormal input to the striatum from the amygdala or midbrain could result in increased somatic motor output and ticking behavior [39,42,43]. Therefore, it is possible that the limbic system can affect emotional responses, in turn being related in some cases to ticking behaviors based on the neural pathways involved. 

### 2.2. Non-Motor Aspects of Tics

Environmental and behavioral factors such as presence or absence of specific stimuli and emotional reactions to life events, different settings or activities have been found to significantly impact tics. The results of a study performed by Findley and colleagues using the Yale Children’s Global Stress Index (YCGSI) showed that not only are tics in children with TS and obsessive–compulsive disorder (OCD) affected by emotional stress, but these children experience more of these stressors in their daily lives [10]. Numerous studies have noted that physical or psychological stress, anxiety, tension, and frustration are associated with an increase in tic expression [14]. Factors that affect ticking behaviors can be thought of as antecedent or consequential, happening before or after a tic occurs, respectively.

PSPs are an extremely important sensory antecedent of ticking behavior. The premonitory urge, a type of PSP, is associated with inner tension or discomfort that is thought to be relieved by ticking. Many individuals have reported that their ticking behavior is primarily associated with the alleviation of discomfort associated with premonitory urges, and tic suppression intensifies these premonitory urges [17]. These phenomena originate in the cortico-striatal-thalamo-cortical circuit, which as discussed above, is heavily involved in tic expression. Current explanations of PSPs are related to an oversensitivity to stimuli due to excessive stimulation of the sensorimotor cortex [17]. Pogorelov and colleagues [44] found that injecting a GABA antagonist into the sensorimotor cortex leads to increased tics in male rats, suggesting that the disinhibition of the sensorimotor cortex could possibly underlie PSPs due to the association with ticking. Additionally, the negative emotions discussed above are thought to be associated with amygdala activation: excessive output from the amygdala in addition to the cortex can lead to the activation of GABA-A receptors in the striatum, producing tics through the stimulation of pathways in the thalamus and M1 region [17].

Consequent factors happen after a tic occurs and either decrease or reinforce ticking behaviors. Negative consequences, such as telling a child to leave the room or publicly commenting on their tics, were reported to have negative outcomes. Positive consequences, such as rewarding a child for modifying their ticking behavior, had a positive outcome [14]. While conditioning may affect tic expression in the long run, emotional components can have an immediate effect on the outcome of ticking behaviors.

The majority of recent studies agree that tics are mostly exacerbated by high or low sensory stimulation, anxiety-inducing situations, frustration and anger, fatigue and sleep loss. Emotional responses to certain types of anxiety, frustration, and boredom also appear to increase the activation of specific areas of the cortico-striatal-thalamo-cortical circuit, ultimately facilitating the stimulation of the striatum [17]. It is possible that this stimulation is facilitated by the release of dopamine, which may help reduce the discomfort of PSPs and reinforce ticking behaviors. The neurobiology of the stress response, as represented in Figure 3, including the HPA axis, may play a role in this release of dopamine in response to stressors, and therefore the stimulation of the striatum. 

The acute stress response is mediated by the HPA axis, which releases hormones that are essential for coping with stress. This process is initiated by the secretion of the corticotropin-releasing hormone (CRH) from the hypothalamus, which in turn promotes the release of the adrenocorticotropic hormone (ACTH) from the pituitary gland, and cortisol from the adrenal cortex [17]. This rapid surge of hormones is thought to be mediated by the release of dopamine, which has been documented in the prefrontal cortex, nucleus accumbens, and dorsal striatum [17]. Additionally, many studies have observed that individuals with TS have a larger HPA axis activation, producing larger amounts of CRH, ACTH, and cortisol [17]. It is highly likely that there is an HPA axis response to PSPs: from this perspective, it is possible that, “Tics may first occur as a by-product of these mechanisms, and then be progressively consolidated as a maladaptive coping response throughout the developmental trajectory of the disorder” [17]. While HPA axis activation can exacerbate tics due to their origin in the abnormalities of the basal ganglia, these abnormalities may contribute to ticking behaviors in neurological disorders such as Sydenham’s chorea (SC) and in the controversial pediatric autoimmune neuropsychiatric disorder associated with streptococcus (PANDAS).

### 2.3. Expression Networks

Recording neuronal activity before, during, and after tic activity is important for the understanding of the neuronal pathways that affect ticking behaviors. Many studies attempted to explain the pathways of ticking behaviors, which were found to include the motor, sensory, limbic, and executive networks. Wang et al. [45] compared the neural circuits involved in spontaneous tics and tics produced by an individual imitating his/her own ticking behaviors. The investigators found that, “Activity in the sensorimotor cortex, putamen, [globus pallidus] (GP), and SN is greater during the expression of spontaneous tics than in the execution of voluntary movements mimicking them” [45,46] There was observed activation in motor pathways such as in the M1 and putamen, in addition to sensory and limbic areas that preceded a tic in an fMRI study by Neuner and associates (2014) [47]. This same study found an increased activity in the thalamus, M1, and somatosensory cortex during tic production. Overall, it was observed that tic activity is associated with neuronal activity in the motor elements of the basal ganglia-thalamo-cortical system [46].

The neural components of premonitory urges have been investigated through multiple fMRI studies. An increased activity in the paralimbic areas, including the anterior cingulate cortex, insular cortex, and sensory areas, was reported before tic onset in addition to sensory areas such as the parietal operculum [46,47]. This indicates that some of the same networks involved in ticking behaviors are also involved in premonitory urges, which have been reported to be a factor in voluntary ticking behavior [46].

Many individuals with TS experience comorbid conditions, such as ADHD and OCD. The fact that these disorders coincide with TS has led many researchers to think that they share pathways in the brain; indeed, this has been observed in multiple studies. Smaller GP [48], caudate nucleus, and putamen (Nakao, 2011) [49] volumes have been reported in ADHD individuals. Additionally, caudate nucleus (Bloch et al., 2005) [50] and putamen [51] volumes were found to be reduced in children with tic disorders and comorbid OCD compared to tic disorder individuals without comorbid OCD. The caudate nucleus volume in these individuals was negatively correlated with OCD symptoms and tic severity, demonstrating a role of the striatum in both OCD and tics [46,50].

Additionally, there is evidence that the brain of a person with a tic disorder is different from an individual with a neurotypical brain, even in the absence of ticking behaviors. These differences are primarily found in the motor and sensory systems [46]. Many studies observed that the ability to suppress undesired movements, also known as cortical motor inhibition, is thought to be related to ticking behaviors [46]. Another change in the brain of a person with a tic disorder is concerned with sensory hyperawareness, known as introspective awareness, and is thought to correlate with the sensation associated with premonitory urges [52]. Introspective awareness is, “Associated with enhanced activity of the insula, motor, and cingulate cortices” [46,53]. Individuals with tic disorders have also been observed to demonstrate sensory-gating deficits (Swerdlow and Sutherland, 2005) [54] and prepulse inhibition disruption [55].

Many individuals with ticking behaviors self-report that contextual factors can impact tic frequency and severity, including stress and anxiety [14]. However, when Conelea and colleagues [54] compared tic frequency in stress-manipulated situations against baseline levels, they found no significant difference. It was only when individuals were asked to consciously suppress their tics that tic frequency increased, suggesting that stress and other factors may indirectly affect the ability to voluntarily suppress tics [46,56]. It is also possible that tic expression is partially and indirectly controlled by different behavioral states that involve dopamine and noradrenaline, “which may be regulated in turn by additional neuromodulators, such as histamine” [46,57].

## 3. Behavioral Aspects of Tics

Emotional states can involve the asymmetric activation of dopaminergic pathways, with hemispheric activation depending on the amount of time exposed to stress [58]. For example, while Parkinson’s disease is not considered a tic disorder, it is still associated with basal ganglia dysfunction, and it was found that individuals with right-sided (left basal ganglia abnormalities) symptoms, compared to those with left-sided symptoms, were reported to experience more depressive symptoms [58]. Therefore, it is possible that, with regard to ticking behaviors, comorbid conditions could be an indication of hemispheric abnormalities in the basal ganglia that can produce ticking behaviors [59]. 

A variety of emotions can exacerbate ticking behaviors, including frustration, anger, and anxiety. Problems completing complex tasks due to ticking behaviors can lead to a level of frustration associated with the consequence and anticipation of the task performance, especially if the task involves controlled regulation [60] in open-loop systems. In open-loop or non-feedback-based systems, the control action from the controller is not constrained by the “process output”, which is the process variable that is being controlled. It does not use feedback to govern its output depending on whether or not the desired goal is achieved. 

It is possible that frustration can also impair a ticking individual’s ability to plan actions using visuo-spatial cues, which could result in an over-reliance on somatosensory proprioception to know when an action has been accomplished. Because of this, the muscles are over-ready for action, leading to increased muscle tension. By definition, muscles must be tensed in order to tic: many people who report premonitory urges also report high overall muscle tension, based on self-reports [13,60]. Another study by O’Connor and colleagues found that tic onset is predominantly associated with dissatisfying and tension-producing activities, [13,61] thus confirming an association between frustration and tic onset.

A study by Cavanna et al. [62] confirmed that anger also plays a role in ticking behaviors. It was found that individuals with TS scored higher on the Conners’ Parent Rating Scales-Revised (CPRS-R) and Conners’ Teacher Rating Scales-Revised (CTRS-R), with “most scores falling within the ‘borderline’ and ‘pathological’ range for behavioral disorders” [62]. These individuals also scored higher on the Child Behavior Checklist (CBCL) compared to their neurotypical peers. While this study has a limited sample size, and the results should, therefore, be analyzed with caution, other investigations have supported Cavanna and associates’ [62] conclusion that individuals with tic disorders tend to have higher instances of anger and aggression. A study conducted by Freeman [63] found that anger is considered to be a comorbid symptom of tic disorders, especially when ADHD is also present. In fact, it is thought that ADHD accounts for the anger and aggression commonly seen in TS individuals [63].

Anxiety is also thought to play a role in tic disorders, as confirmed by a study conducted by Coffey et al. [64]. General psychiatric comorbidity was overwhelmingly abundant in the 190-person sample size (100% in individuals classified as having severe tics, 94% in individuals classified as having moderate tics), including anxiety disorders. Anxiety disorders (with the exception of simple and social phobia) were severely overrepresented, including “panic disorder, agoraphobia, separation anxiety disorder, and overanxious disorder” [64]. It was observed that separation anxiety most strongly predicted tic severity, and presence of multiple anxiety disorders was thought to be associated with a 3.5× higher likelihood of severe ticking behavior [64]. Along the lines of separation anxiety described above, Dehning and colleagues found that TS individuals showed higher levels of attachment anxiety and attachment avoidance in their relationships, possibly due to maladaptive experiences in childhood [65]. Additionally, a case study of a 6 y/o boy was presented, in which he experienced multiple vocal and motor tics along with symptoms of anxiety related to separation from his mother [66]. However, after cognitive-behavioral therapy and attachment-focused therapeutics, tic frequency eventually decreased, highlighting the importance of addressing comorbid conditions associated with ticking disorders.

There are reports of stress-related fluctuations on tic severity, commonly occurring due to fatigue, emotional trauma, anxiety, and stress [67,68]. A stress response is initiated by the release of CRH from the hypothalamus, which then stimulates the synthesis and release of ACTH from the pituitary gland, which ultimately allows for the release of glucocorticoids (including cortisol) from the adrenal gland [67]. Cortisol is an important homeostatic regulator whose secretion follows a circadian rhythm, with levels being highest in the morning and lowest at night. One of the most widely-used biomarkers of stress is the circulating concentration of cortisol secreted by the adrenal gland, which plays a large role in the HPA axis and in the stress response. A study conducted by Corbett and colleagues [67] examined the diurnal cortisol pattern and reactivity of the HPA axis in children with TS. They found that children with TS displayed increased levels of anxiety across all levels on the Multidimensional Anxiety Scale for Children compared to typically developing children. Additionally, while there was no statistically significant difference in the morning and afternoon diurnal cortisol rhythm between the TS and control groups, it was noted that subjects with TS displayed lower levels of cortisol at night [67]. Because lower evening cortisol profiles are associated with chronic stress conditions [69,70,71], it is thought that these decreased levels in TS individuals could be the result of daily stress, possibly due to daily tic suppression. Additionally, when the participants’ stress responses were tested by placing them in an MRI apparatus, children with TS showed increased cortisol levels in response to the MRI environment when compared to their neurotypical peers [67]. This supports a heightened level of responsiveness of the HPA axis in response to stress in children with TS [67]. 

With increased stress usually comes an increase in heart rate. TS individuals commonly report symptoms consistent with sympathetic nervous system overactivity, including increased heart rate, nervousness, and agitation [68,72]. A study performed by van Dijk and associates [73] found that, during a valsalva test, individuals with TS displayed a larger maximum (but not a minimum) heart rate compared to neurotypical controls, possibly due to the initial heart rate in this group of TS individuals being higher than the control group [72,73]. Additionally, in a study looking at cardiovascular and catecholaminergic activity during mental load, Tulen et al. [74] found enhanced cardiovascular activity in tic individuals, “with higher heart rate and blood pressure during baseline compared to healthy controls” [72,74].

Galvanic skin response (GSR) is another easy way to measure sympathetic autonomic nervous system activity and can reflect changes in peripheral autonomic arousal. A study conducted by Nagai and colleagues [75] observed how changes in sympathetic arousal (induced using GSR biofeedback) impact tic frequency in individuals with TS. The total number of tics was significantly lower during relaxation GSR biofeedback than during arousal GSR biofeedback, reflected in the frequency of facial and motor tics. In the arousal biofeedback condition, there was a significant correlation between GSR activity “and the number of tics at the start of the session such that higher sympathetic tone was associated with increased tic frequency across individuals” [75]. Therefore, it is hypothesized that GSR biofeedback training could help in the treatment of individuals with TS due to the observed lower tick frequency in the relaxation GSR biofeedback condition.

While the GSR is associated with a response to stress, it is also thought that perceived stress by individuals with tic disorders can influence ticking behavior. A study performed by Lin and colleagues [76] examined the relationship between psychosocial stress and tic fluctuations in children and adolescents with TS and/or OCD. Psychosocial stress was measured by a participant’s self-report, parental report, and clinician ratings of long-term contextual threat. Overall stress levels were found to be higher in TS and OCD individuals, but ticking behaviors tended to decrease. However, even after controlling for age, Lin and colleagues found that the participant’s current levels of psychosocial stress and depression were an independent but significant predictor of future tic severity [76].

## 4. Conclusions

Overall, the basal ganglia is a vast network that reaches many important areas of the brain. Dysfunction in this system (namely in the GABAergic networks) can create an excess of striatal dopamine, which excites the basal ganglia-thalamo-cortical system, producing an excess of motor output (ie. motor tics). The limbic territory of the basal ganglia, including the ventral striatum, is thought to induce verbal tics. Because the limbic system and its structures are involved in emotional processing, it is thought that external stressors can exacerbate abnormalities along this pathway, influencing tic output. Additional dopamine produced by the HPA axis during stressful situations further increases the disinhibition of this system, leading to an increase in ticking behaviors. The literature described above provides evidence of external factors relating to anxiety, tension, stress, and frustration leading to an increased incidence of tic expression and possible indication of future tic severity. This is highly influenced by PSPs, which are so uncomfortable that individuals will voluntarily tic to rid themselves of the discomfort. It is also likely that stress and other environmental factors can influence an individual’s ability to suppress tics.

There exists an additional immune component to ticking behaviors with regard to the stress response via the HPA axis. Stress can indirectly lead to an overactive immune response. It is important to note that while there is strong evidence to support the HPA axis hypothesis in tic behavior post trauma, this relationship has not yet been solidly associated in genetic studies of TS and other tic disorders.

Additionally, comorbidities such as OCD and ADHD are associated with tics, and likely utilize similar pathways in the basal ganglia-thalamo-cortical system. While ticking behaviors may seem simple on the surface, their underlying pathophysiology and the extensive networks that contribute to this behavior are highly complex and are likely influenced by everyday stressors that connect emotional state to ticking behaviors.

## Figures and Tables

**Figure 1 brainsci-12-00242-f001:**
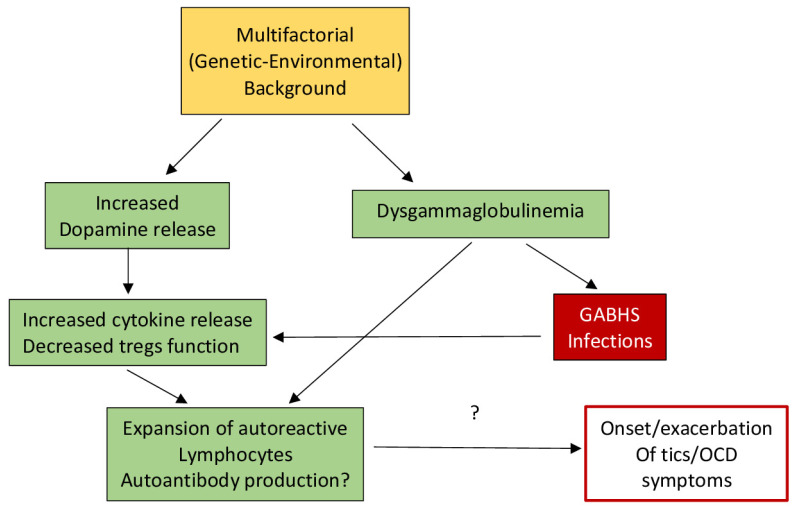
Hypothetical model of the basis for dysfunction in TS. Secondary to the reduction in T regulatory lymphocytes (Tregs), an enhancement of cytokines and reduction in immune tolerance may be facilitated by increased dopamine levels as well as dysgammaglobulinemia, which could possibly facilitate autoimmunity (after Eelamin, I., Edwards, M. J., and Martino, D. (2013). Immune dysfunction in Tourette syndrome. *Behavioural neurology*, *27*(1), 23–32. with permission).

**Figure 2 brainsci-12-00242-f002:**
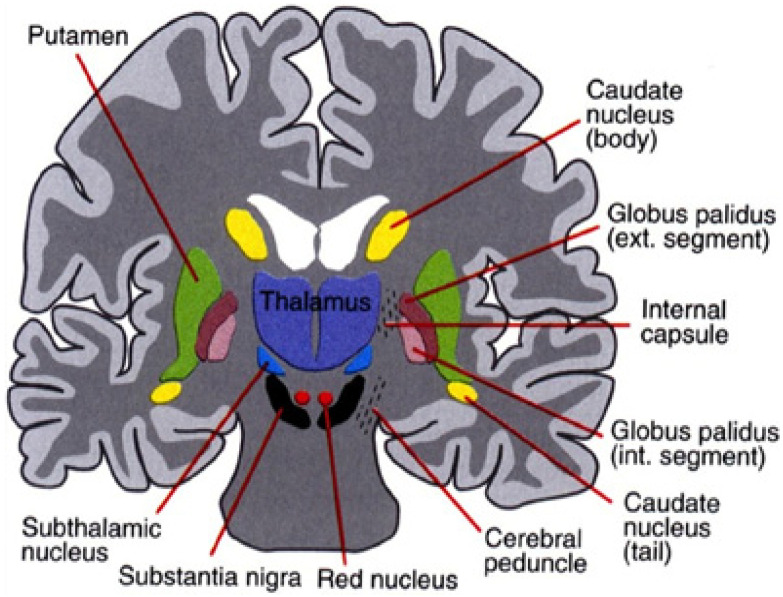
The basal ganglia that clinically includes sub-thalamic nucleus and substantia nigra whose component structures are highly interconnected (after Leisman, G., Braun-Benjamin, O., and Melillo, R. (2014). Cognitive-motor interactions of the basal ganglia in development. *Frontiers in systems neuroscience*, *8*, 16. with permission).

**Figure 3 brainsci-12-00242-f003:**
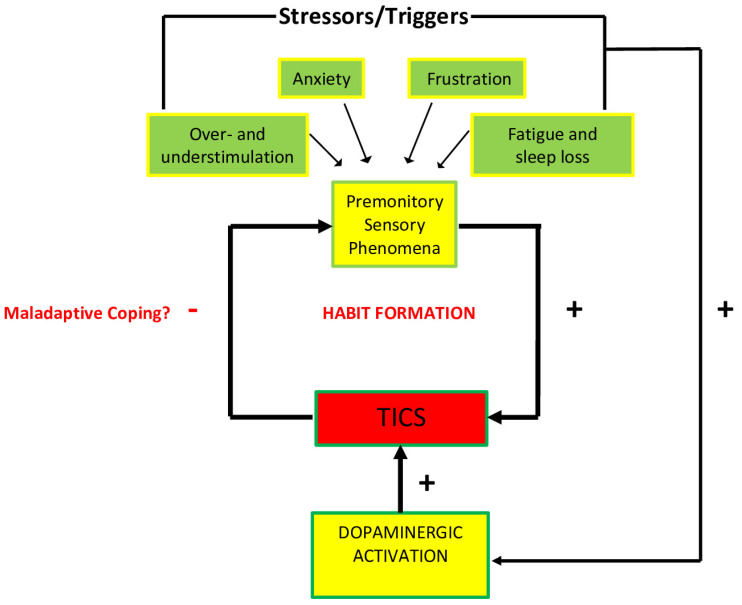
Diagrammatic representation of the association between promontory sensations and ticking behavior. Environmental and contextual triggers intensify PSPs, as well as dopaminergic activity that are hypothesized to increase tic behavior. The tics themselves affect the intensity of PSPs that could in turn potentially create a maladaptive habit as a coping mechanism (after Godar, S. C., and Bortolato, M. (2017). What makes you tic? Translational approaches to study the role of stress and contextual triggers in Tourette syndrome. *Neuroscience & Biobehavioral Reviews, 76*, 123–133. doi:10.1016/j.neubiorev.2016.10 (2017), with permission).

## Data Availability

Not applicable.

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
