# Peer review of "Tics and Emotions"

_brainsci, 2022, doi:10.3390/brainsci12020242_

Round 1

Reviewer 1 Report

Tics and stereotypies are two repetitive and involuntary movement disorders, initiated in childhood. Both movements have different motor pattern, but they are influenced by emotions.

It would be important to have a new review for stereotypies and emotions in the same way

Author Response

Tics and stereotypies are two repetitive and involuntary movement disorders, initiated in childhood. Both movements have a different motor pattern, but they are influenced by emotions.

We thank the reviewer for the helpful comment and agree that tics and stereotypies are related, mostly but not necessarily, in the case of tics, initiated in childhood. There have been a number of good reviews on the topic and their conclusions have been added and integrated within the introductory section of the manuscript.

These reviews have included:

Harris KM, Mahone EM, Singer HS. Nonautistic motor stereotypies: clinical features and longitudinal follow-up. Pediatric neurology. 2008 Apr 1;38(4):267-72.

Martino D, Hedderly T. Tics and stereotypies: A comparative clinical review. Parkinsonism & Related Disorders. 2019 Feb 1;59:117-24.

Robakis D. How much do we know about adult-onset primary tics? Prevalence, epidemiology, and clinical features. Tremor and Other Hyperkinetic Movements. 2017;7.

The findings are incorporated into the updated typescript.

It would be important to have a new review for stereotypies and emotions in the same way.

We likewise agree with the reviewer that such an updated comparison between the neurophysiology of tics and stereotypies would be valuable and the instant paper focuses on tics in general, its limbic and basal ganglia connectivities, and the HPA-Axis.

Reviewer 2 Report

The review provided with this manuscript deals with a relevant topic and gives an interesting overview from the perspective of cerebral circuitry.

The authors did a good work in summarizing existing literature and in putting it in a framework. However, I have three points to raise.

1) The first sentence of the manuscript states that "Tics are a phenomenon characteristic of many neurological disorders". One could dispute if tics are a phenomenon or a symptom, but a taxonomy of neurological disorders having tics as a clinical feature is needed following this opening. Moreover, given the differences between these disorders, the authors should discuss the characteristics of tics in the specific context (i.e. tics in TS are not tics in PANDAS).

2) No differentiation is provided according to age of subjects. This is however a potentially relevant factor. In fact, most of the existing studies concerning children and adolescents are ignored in this review. (see e.g. studies by Cavanna & his research group).

3) It is unclear if the authors consider psychiatric comorbidities as a consequence of the biological alteration of the basal ganglia and of their connections, or if the presence of psychiatric comorbidities is a factor that can differentiate different conditions having tics as a feature.

Author Response

The review provided with this manuscript deals with a relevant topic and gives an interesting overview from the perspective of cerebral circuitry.

The authors did good work in summarizing existing literature and in putting it in a framework. However, I have three points to raise.

We thank the reviewer for the vote of confidence

1) The first sentence of the manuscript states that "Tics are a phenomenon characteristic of many neurological disorders". One could dispute if tics are a phenomenon or a symptom, but a taxonomy of neurological disorders having tics as a clinical feature is needed following this opening. Moreover, given the differences between these disorders, the authors should discuss the characteristics of tics in the specific context (i.e. tics in TS are not tics in PANDAS).

We agree with the reviewer's comment that ticking may be associated with neurological disorders or a symptom of some other process. We have adjusted the opening sentence accordingly and have distinguished between tics and stereotypies in the same introductory paragraph, making reference to both the frequent childhood-onset v. adult-onset.

We had wanted to keep the focus on the relation between ticking behavior, the limbic system, basal ganglia, and the HPA-Axis. Others have done a reasonable job of dealing with these taxonomic issues and three papers, in particular, that should have been referenced have been added in this context and they included:

Harris KM, Mahone EM, Singer HS. Nonautistic motor stereotypies: clinical features and longitudinal follow-up. Pediatric neurology. 2008 Apr 1;38(4):267-72.

Martino D, Hedderly T. Tics and stereotypies: A comparative clinical review. Parkinsonism & Related Disorders. 2019 Feb 1;59:117-24.

Robakis D. How much do we know about adult-onset primary tics? Prevalence, epidemiology, and clinical features. Tremor and Other Hyperkinetic Movements. 2017;7.

2) No differentiation is provided according to age of subjects. This is however a potentially relevant factor. In fact, most of the existing studies concerning children and adolescents are ignored in this review. (see e.g. studies by Cavanna & his research group).

We have included references to Cavanna and his group in references [2], [52], [62], [72], and [75] albeit not stating that we are referring to childhood. we think that the opening statement of tic onset largely, although not exclusively being of childhood-onset, answers that question. In addition, an important study by Bloch and colleagues of caudate nucleus and putamen on comorbidities with OCD and ticking behavior has been included associated with reference [50].

3) It is unclear if the authors consider psychiatric comorbidities as a consequence of the biological alteration of the basal ganglia and of their connections, or if the presence of psychiatric comorbidities is a factor that can differentiate different conditions having tics as a feature.

Again, the reviewer raises an important point relating to the issue of psychiatric comorbidities. The focus of the paper is the relation between ticking behavior given that the purpose of the paper was the linkage between tics and emotions, in particular with reference to the HPA-AXIS (stress), the basal ganglia and by extension the role of the direct and indirect pathways connectivities with the frontal lobe's motor and cognitive functions, particular comorbidities are of necessity discussed in particular relating to OCD. However, given that we had brought autoimmunity into the discussion, we have referenced, albeit not in-depth, the function of dopamine receptors prevalent in the basal ganglia-thalamocortical-circuit whose function is thought to be impaired with disorders involving the CNS dopaminergic circuits cited in the paragraph of references [27-33]. The potential comorbidities thusly could be associated with schizophrenia and major depression. In addition, Ferrari and colleagues' work has been referenced.

We have additionally noted that ticking behavior's comorbidities include a significantly high comorbid incidence of ADHD as well as OCD now being found in the paragraph with references [48-51] in addition to the inclusion of Bloch and colleagues of the role of the caudate nucleus and putamen in comorbid OCD and ticking behavior [reference 50]

Reviewer 3 Report

This manuscript is a comprehensive review on the effects emotions exert on the HPA axis which in turn eventually gives rise to tics.

It is an in-depth article that includes a detailed description of the anatomical circuitry (motor and non-motor, expression) and the behavioral aspects of tics.

They make their case rather convincingly, based on the growing evidence of involvement of extrinsic factors towards tic onset.

The base of the manuscript is the HPA axis hypothesis, that has not yet been solidly associated in genetic studies with TS and tic disorders, but there is growing evidence towards it. This article will grow increasingly useful for new researchers, as more evidence accumulates for the HPA axis.

Author Response

This manuscript is a comprehensive review on the effects emotions exert on the HPA axis which in turn eventually gives rise to tics.

It is an in-depth article that includes a detailed description of the anatomical circuitry (motor and non-motor, expression) and the behavioral aspects of tics.

They make their case rather convincingly, based on the growing evidence of involvement of extrinsic factors towards tic onset.

The base of the manuscript is the HPA axis hypothesis, which has not yet been solidly associated in genetic studies with TS and tic disorders, but there is growing evidence towards it. This article will grow increasingly useful for new researchers, as more evidence accumulates for the HPA axis.

We thank the reviewer for the kind words and overall positive review. We have included a statement in the manuscript (in the conclusion section) delimiting the HPA-Axis hypothesis as lacking as yet, solid genetic studies but increasing support for the hypothesis.

Round 2

Reviewer 2 Report

I think the authors effectively answered my previous concerns.

I think the manuscript deserves to be published.